# The Role of NRF2 in Cerebrovascular Protection: Implications for Vascular Cognitive Impairment and Dementia (VCID)

**DOI:** 10.3390/ijms25073833

**Published:** 2024-03-29

**Authors:** Yizhou Hu, Feng Zhang, Milos Ikonomovic, Tuo Yang

**Affiliations:** 1Department of Neurology, University of Pittsburgh, Pittsburgh, PA 15216, USA; huy5@upmc.edu (Y.H.); zhanfx2@upmc.edu (F.Z.); ikonomovicmd@upmc.edu (M.I.); 2Pittsburgh Institute of Brain Disorders and Recovery, University of Pittsburgh, Pittsburgh, PA 15216, USA; 3Department of Internal Medicine, University of Pittsburgh Medical Center (UPMC) McKeesport, McKeesport, PA 15132, USA; 4Department of Psychiatry, University of Pittsburgh, Pittsburgh, PA 15216, USA; 5Geriatric Research Education and Clinical Center, VA Pittsburgh Healthcare System, Pittsburgh, PA 15240, USA; 6Department of Internal Medicine, University of Pittsburgh Medical Center (UPMC), Pittsburgh, PA 15216, USA

**Keywords:** aging, blood–brain barrier, oxidative stress, neuroinflammation, perivascular macrophage, lymphatic system

## Abstract

Vascular cognitive impairment and dementia (VCID) represents a broad spectrum of cognitive decline secondary to cerebral vascular aging and injury. It is the second most common type of dementia, and the prevalence continues to increase. Nuclear factor erythroid 2-related factor 2 (NRF2) is enriched in the cerebral vasculature and has diverse roles in metabolic balance, mitochondrial stabilization, redox balance, and anti-inflammation. In this review, we first briefly introduce cerebrovascular aging in VCID and the NRF2 pathway. We then extensively discuss the effects of NRF2 activation in cerebrovascular components such as endothelial cells, vascular smooth muscle cells, pericytes, and perivascular macrophages. Finally, we summarize the clinical potential of NRF2 activators in VCID.

## 1. Introduction

Vascular cognitive impairment and dementia (VCID), an age-related neurodegenerative disorder, is the second most common cause of dementia after Alzheimer’s disease with no available treatment yet [1]. VCID accounts for approximately 20% of dementia cases in the developed world such as North America and Europe, with an even higher percentage in the developing world and Asia [2,3,4]. The prevalence of VCID rises rapidly with increasing age. It is estimated that 1.6% of individuals over 65 years old are suffering from VCID, and the figure increases up to 5.2% among those over 90 years old [2]. In an epidemiological study, the disability-adjusted life year (DALY), a parameter that measures the number of years lost due to illness, disability, or early death, has been estimated as 316 per 100,000 person-years for vascular dementia, and the years of life lived with disability as 85 per 100,000 [5]. It has been estimated that VCID patients have the highest annual cost of care among dementia patients [6]. Hypoperfusion due to vascular dysfunction is the key pathophysiology of VCID [7]. In fact, vascular pathology is fairly common in all types of dementia, and is present in up to 75% of dementia patients upon autopsy [8].

Nuclear factor erythroid 2-related factor 2 (NRF2) plays a crucial protective role in oxidative stress and inflammation [9]. NRF2/Kelch-like ECH-associated protein 1 (Keap1)–antioxidant response element (ARE) signaling is potentially the most important regulation in redox homeostasis [10,11,12]. NRF2 has robust potential for attenuating oxidative and hemodynamic stress in cardiovascular diseases [13]. In this regard, NRF2 is essential for protecting microvasculature, especially when under oxidative stress [14].

In this review article, we first briefly introduce the concept of cerebrovascular aging in VCID and the NRF2 pathway. We then more extensively address the effects of NRF2 in cerebrovascular components such as endothelial cells (ECs), vascular smooth muscle cells (VSMCs), pericytes, and perivascular macrophages (PVMs). Finally, we summarize the clinical potentials of the currently available NRF2 activators in VCID.

## 2. An Overview of Vascular Aging in VCID and the NRF2 Pathway

### 2.1. Cerebrovascular Aging in VCID

The term VCID needs clarification due to the inconsistent and often interchangeable use of the terms vascular dementia (VaD), vascular cognitive impairment (VCI), and vascular cognitive impairment and dementia (VCID), which has been addressed in detail by John et al. [1]. VCI is a recently coined term referring to cognitive impairment caused predominantly by cerebrovascular diseases. It includes a wide spectrum of diseases ranging from mild cognitive impairment to VaD [15]. VaD refers to a broader variant of VCI where disruption to daily activities is present. In the present review, we use the term VCID, which encompasses the full range of disease severity.

Several risk factors have been identified for VCID, with aging and vascular disease considered the most important. Female sex, hypertension, stroke, and brain atrophy on structural magnetic resonance imaging (MRI) also play a role [2,16,17,18,19]. Deterioration of vascular function at different levels of the cerebral vasculature, especially structural components of the blood–brain barrier (BBB) (Figure 1), can occur in normal aging and is associated with dementia [20]. Although accounting for only 2% of body mass, the brain consumes 20% of oxygen and 25% of glucose [21]. Due to the limited ability of the brain to store energy, a constant and sufficient blood supply is essential for the brain to carry out complex and highly energy-consuming tasks [22]. The exchange of nutrients and metabolic waste products occurs predominantly at the capillary level of the brain vasculature [21]. Mounting evidence suggests that aging-related dysfunction of brain microvasculature contributes to neurodegeneration [23,24,25,26].

VCI can be caused by the “entire spectrum of vascular brain pathology” including infarcts, hemorrhages, and diffuse hypoperfusion [27]. The pathobiology of VCID is highly complex, and most commonly involves brain infarcts and white matter (WM) injury [28,29]. Cerebral infarction due to large vessel atherosclerosis and arteriolosclerosis with microbleeds both contribute to the development of VCID [1], with direct tissue injury playing a major role. At autopsy, up to 80% of patients with dementia demonstrate evidence of vascular pathology [30]. Macrovascular infarcts are historically recognized as the major pathophysiology of VCID and are associated with a higher risk of dementia [31]. On the other hand, microvascular infarcts are more common in VCID patients [27,32]; they are present in 20–40% of individuals aged 80 years or older [33] and in nearly half (48.0%) of dementia patients at autopsy [34]. The reasons why microvascular infarcts can cause cognitive impairment are not fully understood, but may involve inappropriate immune response and impairments in waste protein removal [35,36,37] and cerebral hypoperfusion [38]. According to Costantino et al., there are multiple mechanisms that can explain the pathophysiology of cognitive impairment in diseases such as VCID, including WM infarction, BBB breakdown, oxidative stress, inflammation, trophic uncoupling, demyelination and remyelination, neurovascular coupling dysfunction, and disposal of unwanted proteins. The current review focuses on several of these mechanisms to address how NRF2 can play a role in fighting against VCID.

In addition to gray matter injury, WM injury can negatively affect the cognitive function of the brain, as the latter is critical for fidelity and precision of information transfer [39]. Compared to gray matter, WM is more vulnerable to ischemia insult because of its physiologically lower perfusion and grey matter steal [40]. WM lesions are very common in VCID, and their presentation on MRI scans in the form of WM hyperintensities is regarded as the hallmark of small vessel diseases [41,42,43].

Aging is inevitable and irreversible [44]. As an organ mostly composed of postmitotic cells such as neurons and oligodendrocytes, the brain is especially vulnerable to DNA damage which increases and accumulates with advanced aging [44,45]. The association between neuroinflammation and neurodegeneration has been consistently reported. Chronic inflammation, primarily in response to DNA damage, produces excessive reactive oxygen species (ROS) and accelerates the process of age-related neurodegenerative diseases and aging itself [44,46,47]. On top of that, mitochondrial dysfunction is one of the antagonistic responses to DNA damage. Although initially compensatory, the overwhelming ROS production beyond the mitochondrial compensatory capacity eventually increases NF-κB signaling and causes chronic inflammation, which leads to further damage and tissue degeneration [44,48]. Indeed, in a recent VCID study imperatorin was found to reduce mitochondrial membrane potential and thereby ameliorate cobalt chloride-mediated primary hippocampal neuronal damage dependent on the NRF2 signaling pathway [49].

### 2.2. An Overview of the NRF2 Pathway and Its Implications in VCID

NRF2 plays an essential role in the regulation of redox hemostasis and protection of endothelial cells, potentially by targeting ARE-regulated antioxidants through a variety of downstream players such as heme oxygenase-1 (HO-1) and NAD(P)H quinone dehydrogenase 1 [10,50]. Keap1 is the major regulator controlling the activation of NRF2/ARE (Figure 2). Details on the NRF2/Keap1 pathway and its regulation have been thoroughly reviewed by us and other groups elsewhere [51,52,53,54]. In brief, in non-stress conditions, NRF2 is sequestered by the Cul3-Keap1 complex in the cytosol and subjected to proteasomal degradation. NRF2 can be released in response to oxidative stress and subsequently translocated to the nucleus, serving as a transcription factor for downstream antioxidant genes [55,56]. The deactivation of NRf2 is regulated by negative feedback, where NRF2 accumulation induces the expression of the Rbx1-Cul3-Keap1 complex, leading to rapid elimination and deactivation of NRF2 [57]. It is worth mentioning that activation of NRF2/HO-1 also leads to downregulation of proinflammatory cytokines and ameliorates systemic inflammation [58]. The interplay between NRF2 and nuclear factor (NF)-κB tunes the equilibrium of oxidation and inflammation [59,60]. NRF2 regulates inflammation through downregulation of proinflammatory gene expression such as interleukin (IL)-1β and IL-6 and upregulation of anti-inflammatory factors such as HO-1.

NRF2 activity is positively related to species longevity, and NRF2 dysfunction plays an essential role in aging [61]. Indeed, multiple age-related transcriptomic changes, such as proteinopathy, have been observed in NRF2-knockout mice [47]. Accumulating evidence suggests that NRF2 expression and activity decrease in an age-dependent manner [62,63,64]. Similar results were reported in rat and Rhesus monkey aorta organoid cultures [65,66]. In rat livers, NRF2 expression has been confirmed to decrease in an age-dependent manner [62], while the opposite has been observed in the brain, liver, and lungs of young mice [67]. Although NRF2 expression may increase or decrease with age, studies have consistently reported declining efficiency of NRF2 signaling with aging, which is related to decreased antioxidant response [68]. A possible explanation for this is that an age-dependent increase in NRF2 protein levels serves as an internal compensatory mechanism in early aging, while later on a decline in NRF2 expression ensues. In fact, given the multifaceted involvement of the NRF2/KEAP1 pathway in aging physiology and disease pathology, this pathway is currently an emerging hot topic in many fields, including but not limited to metabolic disorders, degenerative disorders, inflammatory disorders, and cancerous disorders [69,70,71,72]. Therefore, it is reasonable to propose that activating NRF2 may be a potential approach to managing age-related and vascular-oriented neurodegenerative disorders such as VCID.

Aging is in a close relationship with vascular dysfunction in both primates and rodents [65,66]. Chronic inflammation has been observed in normal aging, leading to damage and microvascular barrier dysfunction, which might in turn accelerate brain aging [73], forming a vicious cycle. Because NRF2 plays a role in BBB preservation [59] and mitigates inflammation through complex interactions with nuclear factor (NF)-κB signaling, NRF2 may be a promising target for vascular protection in VCID [59]. In addition, the aging process can potentially be slowed down by NAD-dependent enzymes such as SIRT1 that serve to protect the endothelium from oxidative stress [74]. In light of the complex interconnection between NRF2 and SIRT1, it is highly likely that targeting NRF2 might also mitigate vascular aging indirectly through regulating SIRT1 signaling [75,76,77]. The following review of NRF2 functions in different cell types in the brain vasculature is intended to provide a better understanding of its protective role in VCID (Figure 3).

## 3. Effects of NRF2 in Vasculature Components

### 3.1. Endothelial Cells (ECs)

ECs line the lumen of blood vessels and play multiple roles in permeability, inflammation, thrombosis, and regulation of blood flow [78]. In particular, cerebrovascular ECs line the border between the central nervous system (CNS) and the periphery, playing important roles in BBB function and neuroinflammation [79]. One of the earliest manifestations of vascular aging is impaired endothelial-dependent vasodilation function [80]. ECs regulate blood flow by releasing vasodilators such as nitric oxide (NO), and dysfunction of this process plays a major role in the pathophysiology of VCID.

#### 3.1.1. Role of ECs in Inflammation and Oxidative Stress

Inflammation and oxidative stress are important in the pathophysiology of VCID, where aging alone leads to low-grade inflammation [59]. The inflammatory cascade starts with proinflammatory cytokines such as tumor necrosis factor (TNF)-α and IL-1β [60], which subsequently increase ROS production (serving as a major source of oxidative stress [10]) and represent the hallmark of vascular aging [20,59]. ECs play a pivotal role in the inflammatory processes. ECs per se can be activated by proinflammatory molecules such as TNF-α and IL-1, and respond by producing NO and prostaglandin I_2_ which increase the leakiness of venules [81]. ECs can also respond to chronic inflammation with angiogenesis [82,83,84]. In addition, leukocyte and macromolecule extravasation during inflammation penetrates ECs either intracellularly or transcellularly through EC junction proteins [85]

ROS are produced through complex enzyme-mediated processes during aging and cerebral hypoperfusion, and mitochondria are considered their major source [86,87,88,89,90]. Alteration of mitochondrial metabolism causes excessive generation of ROS due to interruption of the electron transport chain [91]. Recent studies have demonstrated that NRF2-Keap1 senses the ROS produced by mitochondria, and it has been suggested that NRF2 plays a role in regulating mitochondrial ROS [92]. Indeed, upregulation of HO-1, a downstream effector of NRF2, has been found to prevent H_2_O_2_-mediated cytotoxicity [50].

NFR2 is related to decreased EC inflammation as well. For example, overexpression of NRF2 through adenoviral transduction significantly decreased expression levels of proinflammatory cytokines, including TNF-α and vascular cell adhesion molecule-1 (VCAM-1), and remarkably increased the expression of antioxidant proteins such as HO-1 and glutathione in aortic EC cultures [93]. NRF2 improved brain EC permeability by reducing inflammation in an in vitro model [94]. As a downstream effector of NRF2, targeted expression of HO-1 demonstrated atheroprotective effects in low-density lipoprotein-receptor knockout mice [95].

#### 3.1.2. Cell Adhesion Molecules (CAMs)

During chronic hypoperfusion and VCID, CAMs are upregulated on EC surfaces to facilitate white blood cell extravasation [96,97,98]. Proinflammatory cytokines such as TNF-α and IL-1β induce upregulation of CAMs such as intracellular cell adhesion molecule-1 (ICAM-1) and VCAM-1 on ECs [99,100]. Therefore, CAMs are important molecules for leukocyte recruitment, which generates oxidative stress and inflammation.

NRF2 may play a protective role in decreasing CAM expression induced by TNF-α [101]. In a mouse model of brain ischemia, NRF2 was found to confer protection by suppressing CAM upregulation in brain EC [102]. In traumatic brain injury (TBI), NRF2 knockdown was found to activate NF-κB and induce proinflammatory cytokines secretion, and resulted in overexpression of CAMs such as ICAL-1 [103]. In human umbilical vein ECs, hydroxyanthranilic acid was found to upregulate NRF2 translocation and subsequently induce HO-1 expression in association with reduced VCAM-1 expression [104]. In cancer studies, NRF2 was found to downregulate the expression of E-cadherin [105]. These findings demonstrate the universal and crucial role that NRF2 plays in regulating CAMs and CAM-induced inflammation.

#### 3.1.3. Neurovascular Coupling (NVC)

Blood flow control is regulated by different cells in different blood vessels. In large arteries, lumen diameter control is endothelial-dependent [106,107]. In small arteries and arterioles, vascular smooth muscle cells control the lumen diameter [108]. The term NVC describes the phenomenon that increasing neural activity leads to increasing local blood supply [109]. EC-dependent NO synthesis and release serve as the predominant modulator of local cerebral blood flow (CBF) for normal NVC functioning [110].

NO synthase (NOS) is critical for matching regional CBF with neural activities, as NOS inhibitors can reduce NVC response by 30% in humans [111]. In addition, endothelial NOS (eNOS) activity is essential for EC proliferation, pericyte recruitment, and angiogenesis after ischemia injury in vivo [112,113]. Aging is related to decreased levels of NO and antioxidant enzymes, and impaired vasodilation is a typical manifestation of aging [20,114,115,116,117]. In addition, chronic brain hypoperfusion is associated with EC dysfunction leading to NVC failure, which is also responsible for WM lesions [97,98]. Impaired NVC is a key pathobiology in neurodegenerations such as VCID [38,118,119,120]. Therefore, maintenance and restoration of normal EC functioning is critical for NVC and VCID management.

EC dysfunction and chronic hypoperfusion are closely associated with oxidative stress, and are modulated by the balance between ROS and NO [10]. ROS inhibits the activity of eNOS, decreasing the level of NO and eventually compromising EC-mediated vasodilation [121]. As we previously described, NRF2 plays a vital role in facilitating NVC by modulating eNOS and tetrahydrobiopterin [59].

It is worth noting that vasodilation is one of the primary pathobiological events during tissue inflammation, and NO is the primary vasorelaxant [83,122]. NO directly facilitates acute inflammation by vasodilation, increasing endothelial permeability and angiogenesis [83]; therefore, it is recognized as a marker of acute inflammation. Proper control of NO levels to allow appropriate vasodilation without compromising its integrity is critical for maintaining normal NVC function.

#### 3.1.4. Junction Protein and BBB

BBB breakdown has been observed in numerous neurodegenerative diseases, including VCID, and may contribute to early stages of cognitive dysfunction [123,124]. The breakdown of BBB leads to neurotoxic chemicals entering the brain and tissue inflammation arising from leukocyte extravasation into the brain parenchyma [125,126,127]. Junction proteins such as the adherens junction (AJ) and tight junction (TJ) are critical for BBB function and participate in regulating the paracellular permeability for the diffusion of water, ions, and small molecules [59,128,129]. The production of TJ proteins relies on normal mitochondrial and NVU function, both of which are vulnerable to oxidative stress [130,131,132,133]. Indeed, decreased expression of claudin-5 protein, the most enriched TJ protein, is observed in ECs with NRF2-knockdown [134,135].

Chronic inflammation and atherosclerosis, which typically accompany VCID, cause excessive blood coagulation, which leads to EC damage and BBB disruption [136]. Subsequent release of cytokines, including vascular endothelial growth factor (VEGF), can further disrupt cerebral vasculature, forming a vicious cycle [137,138]. Chronic cerebral hypoperfusion is directly related to TJ loss, as the release of proinflammatory cytokines such as TNF-α can upregulate the apoptosis signaling–regulating kinase 1 (ASK1), leading to angiotensin II-related EC apoptosis and degeneration of TJ proteins such as claudin and occludin [139,140]. This process could contribute to VCI, as it has been shown that inhibiting ASK1 activation can alleviate memory loss in VaD mice [141].

NRF2 is important for BBB integrity. In vitro models have demonstrated BBB disruption in NRF2 knockout samples, along with reduced expression of junction proteins such as occludin and claudin-5 [134,142]. Through promoter analysis, NRF2 seems to be able to directly target the claudin-5 gene and contribute to stable BBB integrity [134]. Interestingly, obese mice lacking NRF2 have a higher degree of BBB disruption along with a significant increase in blood protein leakage, which is associated with a higher level of oxidative stress [14]. Sulforaphane (Sfn), a potent NRF2 activator, has demonstrated a protective effect against claudin-5 loss through an NRF2-dependent pathway in VCID mice [134,143]. Sfn has proven effective in protecting BBB in stroke and TBI models [144]. Importantly, such a protective effect of Sfn is lacking in NRF2 knockdown mice, supporting a pivotal role of NRF2 in the protection of the BBB [143,145]. Mechanistically, NRF2 directly acts on the promoters of the claudin-5 and VE-cadherin genes, as evidenced by promoter analysis and promoter activity studies [134,144].

Interestingly, in a study with a VCID rat model, oral administration of *Artemisia annua Linné*, a plant harboring antioxidant properties, helped to maintain BBB integrity by increasing platelet-derived growth factor (PDGF) receptor β and platelet-endothelial CAM-1 levels associated with NRF2/Keap1 pathway [145], although CAMs are generally believed to be inflammatory markers and may lead to BBB breakdown.

#### 3.1.5. Cell Death and Angiogenesis

NRF2 dysfunction results in increased ROS-mediated EC apoptosis [65,66,146]. Aging-associated NRF2 dysfunction causes significant endothelial apoptosis and vascular rarefaction [65,147,148,149,150,151]. Without a properly functioning NRF2, ECs lose their protection against physiological ROS production, becoming more vulnerable to pro-apoptotic factors such as H_2_O_2_ or high glucose, as observed in an obesity mouse model [152].

Μitochondrial protection is another important mechanism in NRF2 antiapoptotic function. NRF2 takes part in mitophagy by upregulating PINK1 or p62 expression. NRF2 also plays a role in mitochondrial biogenesis, which compensates for the natural loss of mitochondria, by modulating the expression of related mitochondrial genes [153]. As the upstream gene of NRF2, PI3K/Akt plays an important role during oxidative stress by upregulating NRF-2/HO-1 expression. On the other hand, extracellular regulated kinase (ERK) works directly on the dissociation of NRF2 and Keap1 by phosphorylation of NRF2, which promotes the transcription of NRF2 and thereby upregulates HO-1 expression. PI3/Akt and ERK work together to tune the expression of NRF2. Hypoxia activates PI3K/AKT, while HIF-1α is regulated by ERK1/2. In a lung tissue injury model induced by cerebral ischemia/reperfusion in rats, PI3K and ERK levels were significantly higher compared to the sham group and trended consistently with NRF2/HO-1 [154]. In another study using a rat ischemia/reperfusion model, brain endothelial cells responded to oxidative stress with increased NRF2/HO-1 levels through ERK and Akt phosphorylation [155].

Angiogenesis is an important rescue pathway under the circumstances of hypoperfusion. Evidence suggests that angiogenesis occurs at the same time as neurogenesis and synaptogenesis after ischemic brain injury [156,157]. In angiogenesis, angiogenic growth factors such as VEGF and insulin-like growth factor activate NRF2 at the level of both expression and transcription, the latter of which is likely achieved by activating Akt1 [158,159]. NRF2, on the other hand, promotes angiogenesis by regulating the serine/threonine domain of growth factor receptor [160]. ROS act as a double-edged sword for endothelial angiogenesis [161]. At the physiological level, ROS are essential, and may have a beneficial effect during the tissue repair process by regulating angiogenesis [162]. Acting as signaling molecules, ROS take part in angiogenesis and maintain newly formed blood vessels [163,164,165]. Mechanistically, ROS upregulate angiogenic factors such as VEGF and promote the response of these factors by upregulating their receptors [166,167,168]. On the other hand, pathological levels of ROS may exceed the antioxidant capacity and result in cell death. Moreover, excessive ROS levels are involved in pathological angiogenesis such as in cancer and atherosclerosis [169].

Studies in different models have consistently proved that NRF2 promotes angiogenesis in many organs. For example, knockdown of NRF2 or overexpression of Keap1 impaired angiogenic processes in human coronary artery EC cultures, with abnormal EC adhesion, migration, and microtubule formation [152]. In cardiac microvascular ECs, NRF2 knockdown was associated with reduced VEGF expression level [170]. In the developing retina, NRF2 deficiency reduced angiogenic sprouting and vascular density [171]. In a rodent model of myocardial infarction, resveratrol, an NRF2 activator, promoted angiogenesis [172,173,174]. In VCID models, resveratrol has shown the ability to improve NVC responses by restoring EC function and subsequently improving cognitive function in aging mice [175]. Mechanistically, NRF2 is critical to preserving mitochondrial structural and functional integrity in ECs [176,177], in particular endothelial progenitor cells [178].

In light of the complicated and dual-directional role of ROS in regulating angiogenesis, the extent to which ROS should be eliminated by NRF2 activity is an important issue to consider. Unfortunately, the vast majority of studies have not addressed this concern. In addition, the time window for manipulating angiogenesis might be a critical issue. For example, in the retinopathy of a prematurity model where retinal vessel obliteration was present due to hyperoxia-mediated oxidative stress, Nrf2 was found to be critical for preserving the retinal vessels in P9 mice, but not in those after P12 [179].

### 3.2. Vascular Smooth Muscle Cell (VSMC)

Impaired CBF contributes to cognitive dysfunction, as decreased baseline CBF has been shown to correlate with the severity of cognitive impairment [180]. Atherosclerosis and arteriosclerosis both contribute to the development of VCID [1]; however, dementia caused solely by vascular pathology is rare, and results mostly from large vessel disease, while small vessel disease accounts for most milder forms of VCID [1].

Contractile VSMC phenotype, as its name suggests, controls blood vessel diameters by relaxation and contraction [181]. The local renin–angiotensin (Ang) system (RAS) is critical in vasoconstriction, while NO plays a pivotal role in vasodilation [182,183]. RAS changes contribute to neurodegenerative diseases such as Parkinson’s disease, while Ang II can promote inflammation by producing ROS [184,185]. Synthetic VSMC phenotype, on the contrary, has lower contractile phenotype-related protein expression but is active in the synthesis of proinflammatory factors and matrix proteins as well as in proteins associated with migration and proliferation [181]. Synthetic VSMCs such as H_2_O_2_ proliferate after ischemic arterial injury mediated by ROS [186], which in turn narrows the lumen, leading to reduced CBF. Additionally, they secrete inflammatory factors such as IL-6, indirectly disrupting BBB integrity [187]. Indeed, oral administration of an ROS scavenger reduced proinflammatory cytokines in synthetic VSMCs in a traumatic carotid injury model [55].

Limited evidence is present on the role of NRF2 in VSMCs in VCID. Ang II could induce abnormal proliferation of VSMCs through NADPH-oxidase-mediated ROS production and inflammation, which has been proven to exacerbate pre-existing vascular damage and accelerate atherosclerosis [188]. NRF2 demonstrates antiproliferative effects and is capable of suppressing VSMC migration through the NOX4/ROS/NRF2 pathway. NRF2 depletion, on the other hand, enhanced ROS-dependent VSMC migration upon PDGF stimulation [189,190]. Another research studying the relationship between NRF2 and abdominal aortic aneurysm (AAA) showed similar results; increased NFR2 degradation by Keap1 overexpression resulted in tremendous VSMC-mediated inflammatory factor expression, which subsequently disrupted the aortic structure and led to AAA formation [191,192].

### 3.3. Pericytes—Oxidative Stress and Microvascular Barrier Dysfunction

Pericytes are closely associated with microvasculature, in line with their distribution in small vessels such as arterioles, capillaries, and venules [193]. Pericytes play an essential role in maintaining BBB integrity and in regulating vascular permeability, angiogenesis, capillary diameters, and blood flow, as well as in the removal of toxic metabolites [194,195].

#### 3.3.1. Barrier Function

The location and contractile nature of pericytes suggests their function in regulating microvascular permeability. Increased vascular permeability in the setting of inflammation mostly occurs at the level of venules, the majority of which are surrounded by pericytes. In an in vitro lung pericyte/EC co-culture study, pericytes were proven to serve as an additional barrier compared to EC alone [73]. Permeability of water and a range of different molecular weight tracers were significantly increased in pericyte-deficient mice [194,196,197], suggesting that pericytes may serve as a means of salvage to prevent a higher degree of leakage when EC dysfunction occurs [73]. Several potential mechanisms have been proposed. First, pericytes participate in BBB maintenance, which is highly dependent on redox balance. In a mouse model of hypoglycemia-mediated cognitive dysfunction, both pericyte loss and BBB leakage were restored by Mito-TEMPO, a mitochondria-targeted antioxidant shown to reduce ROS in pericyte cultures [198]. Indeed, in a mouse traumatic brain injury model, TNFα-related oxidative stress and inflammation were responsible for pericyte loss, BBB damage, and vasogenic edema [199]. Second, pericytes interact with ECs to regulate BBB permeability by regulating BBB-specific gene expression on ECs, such as Glut1 and transferrin receptor [194,200]. Last but not least, pericytes promote astrocyte endfeet attachment and normal polarization, thereby maintaining the normal functioning of the BBB [194]. Given the robust antioxidant and anti-inflammatory effect of NRF2, it is plausible that pericytes could be protective of the BBB in VCID, although more direct evidence is required.

#### 3.3.2. Contractile Function/Cerebral Blood Flow

Whether the contractile function of pericytes plays an important role in CBF is controversial. Though studies in the cortex and retina have proved the contractility of pericytes, it is questionable whether and to what extent this contraction impacts the rCBF [201,202]. It has been reported that, unlike VSMCs, pericytes are unable to contract in response to multiple stimuli, which may be due to their lack of smooth muscle actin [203]. In contrast, other studies have suggested that, as compared to VSMC-dominant penetrating arterioles, the majority of CBF upregulation after neuronal activation takes place at the pericyte-dominant capillary level, supporting a crucial role of pericytes in CBF regulation [204,205,206].

Small vessel disease (SVD) is the most common cause of VCID, and it plays a significant role in stroke and Alzheimer’s disease (AD) [207,208,209]. CBF decrease and amyloid β (Aβ) changes are present in both AD and VCID. In AD, Aβ deposition generates excessive ROS and triggers the release of vasoconstrictive peptides such as endothelin-1, altering pericyte tone and causing vasoconstriction [210,211]. Therefore, it is reasonable to hypothesize that NRF2 may rescue pericyte dysfunction and increase CBF in both AD and VCID.

### 3.4. Perivascular Macrophage

Brain-proximal arterioles and venules are surrounded by perivascular space, which hosts perivascular macrophages (PVMs) [212,213,214,215]. PVMs are specialized myeloid cells that play an active role in the maintenance of BBB integrity and lymphatic drainage under physiologic conditions. Increased PVM numbers are observed in diseases associated with infection, vascular impairment, and amyloid deposition [215].

#### 3.4.1. Inflammation/Oxidative Stress

As the resident macrophages in the CNS, PVMs are an important source of ROS [216]. For example, in spontaneous hypertensive mice, damage to the BBB resulted in Ang II accessing the perivascular space, leading to activation of PVMs which produce excessive ROS and resulting in cognitive impairment [216]. On the other hand, BBB breakdown facilitates penetration of fibrinogen into brain tissue, which turns to fibrin and activates microglia, causing inflammation [217]. Several experimental studies have examined the relationship between fibrin and perivascular macrophages. In a rabbit retina model, perivascular macrophages accumulated around fibrin in the setting of IL-1-induced inflammation [218]. Fibrin in the brain tissue was associated with Aβ deposits and loss of pericytes [217]. NRF2 has great potential to break these pathological cascades by eliminating ROS and rescuing cognitive function in VCID.

Two traditional categories of activated macrophages, M1 (pro-inflammatory) and M2 (anti-inflammatory), have been identified [219], though this concept has been increasingly questioned in the field. Our group has demonstrated the presence of PVM subtypes by examining their specific markers [215]. Importantly, NRF2 plays a role in phenotype shifting in macrophages and microglia. For example, human umbilical cord mesenchymal stem cell-derived extracellular vesicles were found to inhibit M1 differentiation of microglia and thereby decrease inflammation levels in a VCID rat model through the PI3K/AKT/NRF2 pathway [220]. In mice modeling Parkinson’s disease, NRF2 KO was associated with significantly increased proinflammatory M1 cytokine levels and decreased anti-inflammatory M2 cytokine levels [221]. In a stroke model, activated microglia converted to pro-inflammatory M1 and anti-inflammatory M2, acting as double-edged sword. Therefore, it could be important to promote differentiation towards the anti-inflammatory state in patients with stroke [222]. NRF2 can likely facilitate the formation of M2 an anti-inflammatory phenotype in PVMs, which would benefit cognitive function in VCID.

#### 3.4.2. Clearance of Interstitial Fluid (ISF) or Metabolic Waste (e.g., Aβ)

Lymphatic drainage is somewhat unique in the CNS because the brain parenchyma lacks lymphatic vasculature. Details on brain lymphatic pathways have been thoroughly reviewed elsewhere [223]. Importantly, perivascular space serves as an important route for the removal of waste products such as Aβ [224]. In transgenic AD mice, an impaired lymphatic system enhanced Aβ deposition [224]. PVMs contribute to lymphatic clearance in two major pathways: the glymphatic pathway and the intramural perivascular pathway [215].

Interestingly, PVMs appear to be a double-edged sword in the pathogenesis of aging-related proteinopathies such as diseases characterized by the pathological accumulation of Aβ, α-synuclein, and p-Tau [225]. Aβ has been reported to impact neurovascular coupling and contribute to impaired cognition [226]. Perivascular space is a major site for both disposal and accumulation of Aβ [37,227,228,229,230]. Residing in the perivascular space, PVMs are, on the one hand, involved in Aβ clearance through phagocytosis; this lessens Aβ deposition in the brain tissue as well as in the blood vessels in the setting of cerebral amyloid angiopathy [231]. On the other hand, PVMs release large amounts of ROS in response to excessive Aβ [232]; in this way, selective removal of PVMs using clodronate liposomes was able to reduce ROS production, preserve BBB integrity, and protect cognitive function in AD mice and spontaneous hypertension mice [232,233].

NRF2 has been proven to influence Aβ levels in the brain. For example, NRF2 deficiency was associated with upregulated Aβ precursor protein (APP) gene expression in obese mice [14]. When crossed with transgenic AD mice with APP or Tau mutations, NRF2 KO led to increased Aβ and p-Tau, resulting in early-onset cognitive dysfunction [47]. Hexahydrocurcumin was found to decrease Aβ and p-Tau production, which are associated with increased NRF2 activity, and to improve memory function in a VCID rat model [234]. In addition, transcriptomic analysis of NRF2-KO mouse brains revealed pathways replicating those in human aging and AD brains. Persistent activation of proinflammatory microglia and expression of proinflammatory cytokines induced by Aβ play a vital role in cellular senescence and the progression of neurodegenerative diseases. At the same time, the SIRT1/NRF2 pathway is partially blocked and NRF2 translocation is diminished by Aβ stimulation, which was reversed by aspirin in a mouse model [232].

## 4. Future Perspectives on NRF2 in VCID

Though preclinical studies support the potential value of NRF2 in VCID prevention and/or management, clinical trials are still lacking at this time. It should be noted that NRF2 dysfunction occurs with aging, which can impair expected responses upon NRF2 activation [59,235]. This is especially important in VCID, where cerebrovascular aging appears to play a central pathogenetic role. Another point to consider is that BBB permeability may or may not be crucial for drug efficacy, given that the vasculature, rather than the brain parenchyma, could be the major site of drug action in VCID, which greatly expands the candidate pool for clinical trial design. In addition, VCID comorbidities such as hypercapnia may impose an additional layer of complexity on NRF2 expression [236], which is commonly the case in the real world given the demographic characteristics of the VCID population.

Studies have identified that NRF2 plays a protective role against newly identified forms of cell death, such as ferroptosis and pyroptosis. Ferroptosis is characterized by iron accumulation and lipid peroxidation [10,11]. In a VCID rat model, gastrodin has been proven to inhibit ferroptosis and improve memory impairment through the NRF2/Keap1–glutathione peroxidase 4 pathway [11]. Pyroptosis is a form of inflammatory programmed cell death mediated by the NLRP3 inflammasome. Activation of ChemR23, a G-protein-coupled receptor, was observed to inhibit pyroptosis and improve cognitive function via the PI3K/AKT/NRF2 pathway in a chronic cerebral hypoperfusion rat model [12]. These could be novel mechanisms for future NRF2-VCID research.

NRF2 activators have been extensively studied in preclinical settings; however, which compounds to proceed with in VCID clinical trials is hard to determine. Five candidate compounds (curcumin, trichostatin-a, panobinostat, parthenolide, and entinostat) have been identified for AD treatment based on structural similarities, NRF2-diseasome, targeted pathways, and modes [237]. A selected list of NRF2 activators that are currently being trialed in CNS diseases is listed in Table 1. Because no NRF2 activators are currently being trialed in VCID studies, patient selection, drug dose, delivery modalities, timing of administration, and administration protocol need to be developed and optimized for the best therapeutic effects while avoiding potential side effects.

In conclusion, NRF2 is a promising target for the management of VCID in light of its robust protective effects in the cerebral vasculature and its multimodal anti-aging potency. In-depth preclinical and clinical trials are needed for better VCID outcomes.

## Figures and Tables

**Figure 1 ijms-25-03833-f001:**
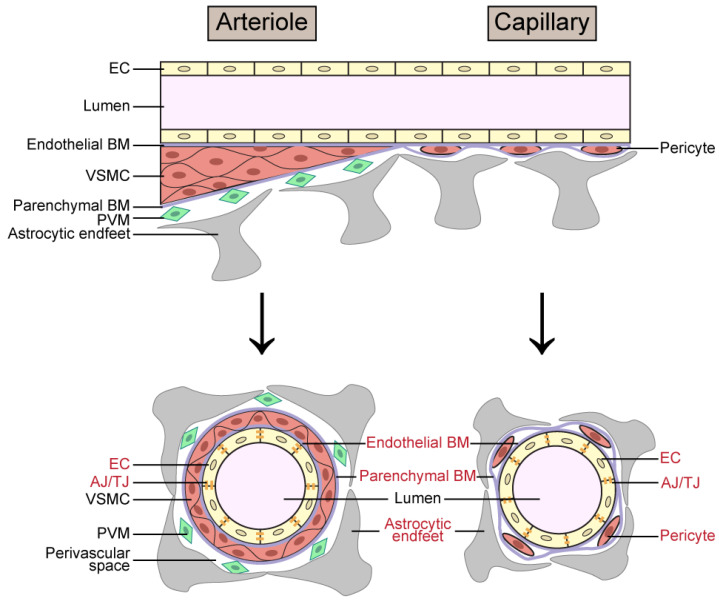
Schematic image showing cell types and structure of the arteriole and capillary/venule levels of the cerebral vasculature. In the lower panel, structural components of the blood–brain barrier are labeled in red. Abbreviations: AJ—adherens junction; BM—basement membrane; EC—endothelial cell; TJ—tight junction; PVM—perivascular macrophage; VSMC—vascular smooth muscle cell.

**Figure 2 ijms-25-03833-f002:**
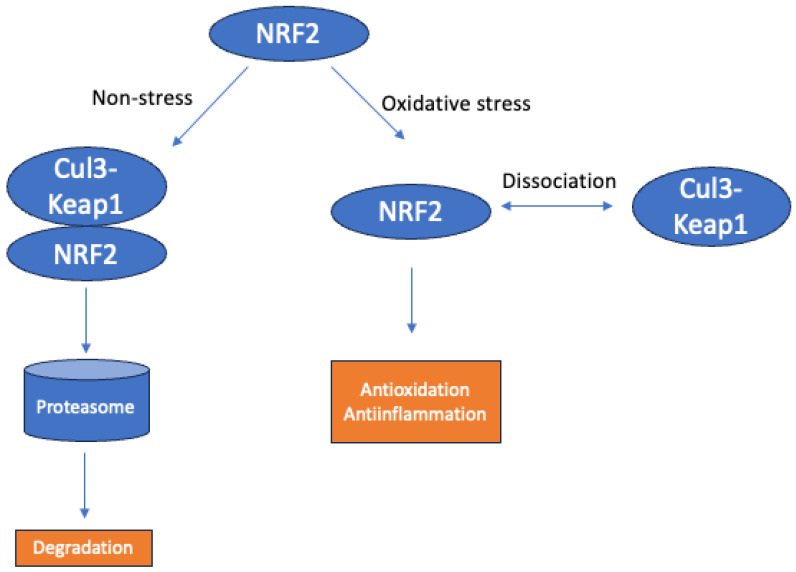
Simplified schematic image of the NRF2/Keap1 pathway.

**Figure 3 ijms-25-03833-f003:**
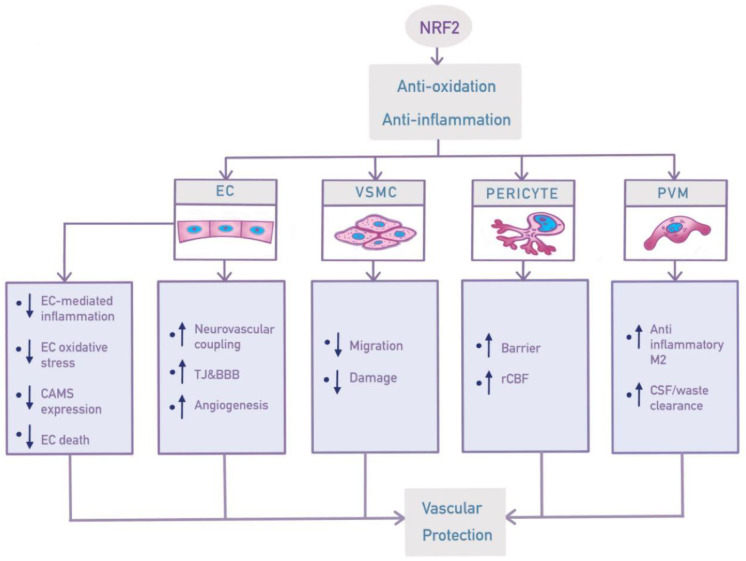
Schematic image showing NRF2 effects in different cell types of the cerebral vasculature in VCID. BBB, blood–brain barrier; CAM, cell adhesion molecule; CSF, cerebrospinal fluid; EC, endothelial cell; PVM, perivascular macrophage; rCBF, regional cerebral blood flow; TJ, tight junction; VSMC, vascular smooth muscle cell. ↑: increase/upregulate. ↓: decrease/downregulate.

**Table 1 ijms-25-03833-t001:** NRF2 activators that are being trialed in CNS diseases.

Compounds	Trial ID	Disease	Comments	References
Dimethyl fumarate	NCT02634307	Multiple sclerosis	FDA approved, currently in phase III trial. Good tolerance. Improvements from baseline in clinical and radiological efficacy outcomes, including significantly reduced annualized relapse rates.	[77]
Omaveloxone	NCT02255435	Friedreich ataxia	Significantly improved neurological function with good tolerance. Mild adverse effects. High accumulation in the brain with significant and stable upregulation of NRF2 downstream genes in monkeys and humans.	[238,239]
Sfn	NCT02561481	Autism spectrum disorder	Only small non-significant changes were found in the primary outcome measures (Ohio Autism Clinical Impressions Scale). Safety and tolerance were confirmed. Significant changes in glutathione redox status, mitochondrial respiration, inflammatory markers, and heat shock proteins.	[240]
SFX-01 (Evgen Pharma)	NCT02614742, NCT01948362, NCT02055716	SAH	Currently in phase II trial. No serious adverse events were reported in healthy volunteers per phase I trials.	[241]

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
