# Peer review of "The Role of NRF2 in Cerebrovascular Protection: Implications for Vascular Cognitive Impairment and Dementia (VCID)"

_ijms, 2024, doi:10.3390/ijms25073833_

Round 1
Reviewer 1 Report
Comments and Suggestions for Authors
The manuscript titled "The Role of NRF2 in Cerebrovascular Protection: Implications for Vascular Cognitive Impairment and Dementia (VICD)" is a thorough review of literature. It is well written and adds to the field of cerebrovascular dysfunction in disease.
Author Response
We thank the reviewer for the positive comment on our manuscript.
Reviewer 2 Report
Comments and Suggestions for Authors
In this manuscript authors briefly introduced cerebrovascular aging in VCID and the NRF2 pathway. Then, they discussed the effects of NRF2 activation in cerebrovascular components such as endothelial cells, vascular smooth muscle cells, pericytes, and perivascular macrophages. Finally, authors summarized the clinical potentials of NRF2 activators in VCID treatment.
The manuscript is interesting and generally well written. However, some points deserve to be improved. In particular:
Minor points
Line 41-45: Since NRF2/KEAP1 signaling is the main topic of this review article, the multifaceted involvement of this pathway deserves to be highlighted. In fact, this pathway is involeved in the onset and progression of several cancerous and non-cancerous diseases (see PMID: 37296665, 37841775 , 37525922, 38314046). The references I mentioned are not mandatory, they are just an example of the multifaceted role of NRF/KEAP1 signalling.
Line 101: authors correctly state that aging is irreversible. However, they should specify that several NAD-dependent enzymes, such as sirt1, can slow this process and are fundamental for good vascular health (PMID: 36829935). Some of these enzymes protect endothelium from oxidative stress, playing a key role in this process (PMID: 34153425).
I suggest to add a table resuming the main results in the studies discussed in the section "3. Effects of Nrf2 in vasculature components"
Abbreviations must be written in full length when mentioned for the first time
Please uniform NRF2 formatting in the text since some times it is written as Nrf2
An accurate revision of typing errors is recommended
Major points
I did not find severe issues to highlight
Author Response
We sincerely thank the reviewer for his/her insightful recommendations, which have been addressed in the revised version. Below please find our point-for-point response to the reviewer’s comments. Text is also revised, with all new edits highlighted in yellow.
Line 41-45: Since NRF2/KEAP1 signaling is the main topic of this review article, the multifaceted involvement of this pathway deserves to be highlighted. In fact, this pathway is involved in the onset and progression of several cancerous and non-cancerous diseases (seen PMID: 37296665, 37841775, 37525922, 38314046). The references I mentioned are not mandatory, they are just an example of the multifaceted role of NRF/KEAP1 signaling.
Response: We thank the reviewer for the valuable suggestion. As recommended, we highlighted the multifaceted involvement of the NRF2/Keap1 pathway in both aging physiology and disease pathology, making it an emerging hot topic including but not limited to vascular dementia, cancerous, metabolic, degenerative, and inflammatory disorders (Line 150-154).
Line 101: authors correctly state that aging is irreversible. However, they should specify that several NAD-dependent enzymes, such as sirt1, can slow this process and are fundamental for good vascular health (PMID: 36829935). Some of these enzymes protect endothelium from oxidative stress, playing a key role in this process (PMID: 34153425).
Response: We thank the reviewer for this great point. We added this point to the revised version of our review, noting that aging could be slowed down by several NAD-dependent enzymes including SIRT1. We also highlighted the complex interactions that exist between NRF2 and SIRT1, the latter might serve as indirectly mechanisms for NRF2’s protection in aging vasculature (Line 162-166).
I suggest to add a table resuming the main results in the studies discussed in the section "3. Effects of Nrf2 in vasculature components"
Response: We thank the reviewer for this suggestion, and we agree that the discussion under Section 3 needs summarization. While revising this article, we did design a table summarizing major results in Section 3. However, we found the content largely overlapped with that of Figure 3, providing unnecessary, redundant information to the readers. Comparing Figure 3 and the table, we finally determined to keep the figure, mainly because 1) figures are more intuitive and readable than tables, and 2) more important, the content under Figure 3 is very straightforward, and readers should not have much difficulty finding a reference if they are specifically interested in any molecular details mentioned in Section 3.
Again, we strongly appreciate this piece of suggestion, and we are open to further discussion/revision.
Abbreviations must be written in full length when mentioned for the first time
Response: We have revised all abbreviations in the text.
Please uniform NRF2 formatting in the text since some times it is written as Nrf2
Response: We apologize for the inconsistency of NRF2 formatting. We have unified them in the revision.
An accurate revision of typing errors is recommended
Response: We have carefully proofread the text in the revised version.
Reviewer 3 Report
Comments and Suggestions for Authors
The manuscript is devoted to the analysis of current scientific data on the role of NRF2 in signaling pathways of mechanisms of vascular cognitive impairment and dementia (VCID). The review also contains brief information about the clinical possibilities of NRF2 activators, which protect nervous tissues and blood vessels from oxidative stress and can be used to protect against VCID.
The information and conclusions in the review are written in a structured and objective manner, and all important and recent scientific papers from the relevant field have been cited.
The work was done at a good methodological level: more than 220 literary sources were analyzed, most of which have been published over the past 10 years.
However, there are several recommendations that will help the authors improve the manuscript:
1. In Figure 1, it is better to indicate in the caption that the right illustration, including schematizes the blood-brain barrier.
2. It seems to me that the manuscript can be supplemented with interesting arguments about the relationship between NRF2 expression, which decreases with hypercapnia (it accompanies hypoxia and cerebral ischemia in VCID). (https://pubmed.ncbi.nlm.nih.gov/28159772/)
Author Response
We sincerely thank the reviewer for his/her insightful recommendations, which have been addressed in the revised version. Below please find our point-for-point response to the reviewer’s comments. Text is also revised, with all new edits highlighted in yellow.
- In Figure 1, it is better to indicate in the caption that the right illustration, including schematizes the blood-brain barrier.
Response: We thank the reviewer for the suggestions on Figure 1. We have added more detailed structures (including 2 layers of basement membranes and tight junction proteins) and highlighted important structures composing the blood-brain barrier.
- It seems to me that the manuscript can be supplemented with interesting arguments about the relationship between NRF2 expression, which decreases with hypercapnia (it accompanies hypoxia and cerebral ischemia in VCID). (https://pubmed.ncbi.nlm.nih.gov/28159772/)
Response: We thank the reviewer for this valuable point. VCID comorbidities are commonly seen in real words which pose an additional layer of complexity. This point has been added in the revision (Line 484-487).
Reviewer 4 Report
Comments and Suggestions for Authors
The paper is very good and complete. It was a pleasure to read. However there are a few observations to be made. these to not concern the scientific quality of the paper, but rather the readability.
figure 1 is completely superfluous. It does not add any essential information or clarifications, it is too big and clumsily made to be of use. Anyway, the level necessary for the reader to understand the paper is much to high for the necessity of such image.
A graph showing the NRF2 pathway could be illustrative, as the paragraph between lines 115-130 is difficult to follow.
the paragraph between lines 475-492 presents the substances which might be used to activate NRF2, but the presentation is too cursory and difficult to follow. Perhaps a bulleted list could be of better use here. Also, as the problem of VCID is very important, maybe an expansion of this chapter would be good, even if the data is limited. The therapeutics are very important here. Also, a graphic aid could be of use.
Author Response
We sincerely thank the reviewer for his/her insightful recommendations, which have been addressed in the revised version. Below please find our point-for-point response to the reviewer’s comments. Text is also revised, with all new edits highlighted in yellow.
figure 1 is completely superfluous. It does not add any essential information or clarifications, it is too big and clumsily made to be of use. Anyway, the level necessary for the reader to understand the paper is much too high for the necessity of such image.
Response: We thank the reviewer for this suggestion, and we agree that the original version of Figure 1 was too superficial for readers of the present article. Instead of deleting it, we revised it in the current version, incorporating more information which is not very well-known and sometimes confusing.
- Two layers of basement membranes are present in the vasculature, both contributing to the healthy BBB.
- We also delivered the point that the BBB is not only present at the capillary level but also at the arteriole level, although the arteriolar level is not typically believed to be the site of material exchange between the blood and the brain tissue.
A graph showing the NRF2 pathway could be illustrative, as the paragraph between lines 115-130 is difficult to follow.
Response: We thank the reviewer for this great point. In the revision, we added a new figure, Figure 2, which briefly illustrates the NRF2 pathway. Molecular details of the pathway is beyond the scope of the present article. we also cited a bunch of articles from our group and others in the text, which are much more thorough in detailed molecular mechanisms, for the reference of readers who are interested (Figure 2 and Line 124-125).
the paragraph between lines 475-492 presents the substances which might be used to activate NRF2, but the presentation is too cursory and difficult to follow. Perhaps a bulleted list could be of better use here. Also, as the problem of VCID is very important, maybe an expansion of this chapter would be good, even if the data is limited. The therapeutics are very important here. Also, a graphic aid could be of use.
Response: We thank the reviewer for this point. We have summarized this paragraph in a table to improve the readability. We also expanded this chapter a bit to provide more information (Table 1 and Line 497-498).